# Catching Common Cold Virus with a Net: Pyridostatin Forms Filaments in Tris Buffer That Trap Viruses—A Novel Antiviral Strategy?

**DOI:** 10.3390/v12070723

**Published:** 2020-07-04

**Authors:** Antonio Real-Hohn, Rong Zhu, Haleh Ganjian, Nahla Ibrahim, Peter Hinterdorfer, Heinrich Kowalski, Dieter Blaas

**Affiliations:** 1Vienna Biocentre, Max Perutz Laboratories, Medical University of Vienna, Center of Med. Biochemistry, Dr. Bohr Gasse 9/3, A-1030 Vienna, Austria; heinrich.kowalski@meduniwien.ac.at (H.K.); dieter.blaas@meduniwien.ac.at (D.B.); 2Institute of Biophysics, Johannes Kepler University Linz, Gruberstr. 40, A-4020 Linz, Austria; rong.zhu@jku.at (R.Z.); peter.hinterdorfer@jku.at (P.H.); 3Department of Laboratory Medicine, Division of Clinical Microbiology, Karolinska Institutet, Alfred Nobels Allé 8/7, 141 52 Huddinge, Sweden; haleh.ganjian@temple.edu; 4Department of Surgery, Surgical Research Laboratory, Medical University of Vienna, Waehringer Guertel 18-20, A-1090 Vienna, Austria; nahla.ibrahim@nyu.edu

**Keywords:** *Enterovirus*, *Rhinovirus*, pyridostatin, filament, extracellular traps

## Abstract

The neutrophil extracellular trap (ET) is a eukaryotic host defense machinery that operates by capturing and concentrating pathogens in a filamentous network manufactured by neutrophils and made of DNA, histones, and many other components. Respiratory virus-induced ETs are involved in tissue damage and impairment of the alveolar–capillary barrier, but they also aid in fending off infection. We found that the small organic compound pyridostatin (PDS) forms somewhat similar fibrillary structures in Tris buffer in a concentration-dependent manner. Common cold viruses promote this process and become entrapped in the network, decreasing their infectivity by about 70% in tissue culture. We propose studying this novel mechanism of virus inhibition for its utility in preventing viral infection.

## Text

Extracellular traps (ETs) are fibrillary networks formed from ~16 nm diameter filaments constituted mainly of nuclear or mitochondrial DNA from mast cells, eosinophils, macrophages, and neutrophils [1]. ETs carry histones, gelatinase, proteinases, elastase, cathepsins, lactoferrin, myeloperoxidase, and other defensins and are part of the innate immune system [2]. ET production is triggered by bacteria, but more recently, viruses were found to also induce ETs [3]. For example, influenza virus infection gives rise to specific ETs that cannot cross-neutralize bacteria but generate inflammation that undermines the alveolar–capillary barrier function and thereby promotes secondary bacterial infection. However, neutrophil ETs also inhibit virus infection [4]. We here demonstrate, via immunolabeling, that a rhinovirus interacts with ETs (Appendix A and Movie S1). Viruses cannot escape from ETs, as this would require nucleic acid hydrolysing enzymes at their surfaces, as are present, for example, in group A *Streptococcus* [5].

A recent in silico study of all available human virus genomes revealed the presence of multiple G-quadruplexes (G4s), repeats of Hoogsteen paired guanosines, in all genomes [6], including those of rhinoviruses, the main cause of the common cold. This prompted us to study whether G4-stabilizing compounds, such as pyridostatin (4-(2-aminoethoxy)-N2,N6-bis[4-(2-aminoethoxy)-2-quinolinyl]-2,6-pyridinedicarboxamide; PDS), might interfere with the release of the ssRNA genome from a prototype rhinovirus, RV-A2, and thus prevent infection via impeding G4 unfolding. PDS and similar compounds are being investigated as anticancer drugs, as they stabilize G4s in telomeres, impacting cellular DNA replication [7]. We found that RV-A2 infection was indeed inhibited upon preincubation with PDS at room temperature. For control purposes, the same incubations of virus and PDS were also carried out at 4 °C, a temperature that very much reduces the diffusion of the compound through the viral protein shell to attain the RNA; capsid breathing dynamically opens conduits in the viral capsid, but this phenomenon is highly temperature-dependent [8]. We were intrigued to see that co-incubation at 4 °C also inhibited infection, but only in Tris buffers. Inhibition of infection upon incubation at room temperature was, however, independent from the buffer and only due to the above G4-stabilization (manuscript under submission). Since PDS contains planar rings (Figure 1a), we wondered whether individual molecules might stack on top of each other and act via aggregating the virus. Such aggregates might bind the virus and reduce its effective concentration. To reveal a putative higher-order structure at an ultrastructural level, 4 µM PDS in water was applied onto freshly cleaved mica and left for 5 min. The PDS solution was then replaced with phosphate-buffered saline (PBS). Using an atomic force microscope (AFM) equipped with a fluidic cell in a Pico-SPM (Molecular Imaging, Phoenix, AZ, USA) [9], we saw that the PDS molecules attached to the mica and aggregated into long fibers with a height of 1.8 ± 0.2 nm (Figure 1b). Such fibers did not form when the PDS was dissolved in PBS (Figure 1c).

The above observations pointed to a dependence of the fiber-generation on the buffer components. To investigate this, we dissolved PDS at 200 µM in different buffers and applied the solutions onto glow-discharged carbon-coated electron microscopy (EM) grids, subjected them to negative staining with phosphotungstate, and observed the samples in a FEI Morgagni 268D electron microscope at 80 kV (Figure 1d). In contrast to the sample adsorbed to mica and observed with AFM, PDS only formed small amorphous adducts in water and very few aggregates (white arrowheads) on the carbon-coated grids. ‘Protofibrils’(black arrows) were also seen with PDS dissolved in Dulbecco’s modified Eagle medium (DMEM, Sigma Aldrich; St. Louis, MO, USA) supplemented with 10% fetal bovine serum (Gibco; Thermo Fisher Scientific, Waltham, MA, USA). However, fibers formed in 50 mM NaCl, 25 mM Tris-HCl (pH 7.5), but not in PBS.

To further study the PDS fiber-generating conditions, we dissolved PDS at various concentrations in the above Tris buffer and observed the samples by TEM (Figure 1e) as above. We noticed a clear concentration dependence of fiber-generation with no fibers occurring at or below 20 µM PDS. ‘Protofibrils’ appeared at 40 µM and well-defined fibers from 60 µM PDS onwards.

We then investigated the influence of 20 and 100 µM PDS in the above Tris buffer on the light-up of SYTO82 (Thermo Fisher Scientific), a fluorescent probe diluted to 5 µM in the same buffer (Figure 1f). Nucleic acids can dramatically increase the fluorescence of such probes due to a forced planarization or rigidification of the probe [10]. For example, production of ETs by different cells has been monitored with SYTOX, a fluorescent probe that lights up upon interaction with nucleotide polymers [11]. The fluorescent signal emission intensity (excitation 541 nm ⁄ emission 560 nm) was acquired at room temperature using a Jasco 6500 fluorometer and plotted as relative fluorescence intensity (RFU). We observed a strong increase of SYTO 82 fluorescence emission upon adding increasing concentrations of PDS. This might be taken to indicate that the PDS can arrange in higher-order structures probably due to π–π stacking and a hydrophobic effect similarly observed in nucleotide polymers (reviewed in Friedman and Honig [12] and references within).

Taken together, the AFM and TEM observation of PDS producing networks of fibers and the increase of the SYTO82 nucleic acid binding capacity led us to ask whether the PDS fibers might be capable of trapping viruses similar to ETs and thereby inhibiting viral infectivity, despite being completely different with respect to their composition. To test that in a physiologically relevant system, we measured the infectivity of PDS-treated virus in HeLa cells. We are aware that such an experiment does not take into account that the fibers might damage the cells similarly to natural ETs (see above); if so, the decreased cell survival could be misinterpreted as increased infectivity. To avoid this, we first decreased the PDS concentrations to 20 µM and incubated 1 µg/mL RV-A2 in Tris buffer for 30 min on ice to prevent capsid breathing and, thus, the interaction of PDS with the viral genome within the protein shell. TEM observation suggested that the presence of virus particles induced the formation of fibers, as they were already observed at the low concentration of 20 µM PDS (Figure 2a). Note that the shape of the PDS fibers differs to some extent in different experiments (left panel).

To test whether the PDS fibers would indeed trap the virus and thus reduce infection, we grew HeLa cells until sub-confluent on coverslips and challenged them with RV-A2, RV-B14, and RV-A89, respectively, at a multiplicity of infection (MOI) of 100, as in Real-Hohn et al. [13]; the viruses had been diluted in Tris buffer with 20 µM PDS as above and incubated on ice for 30 min. For control, the viruses were incubated on ice for 30 min in the same buffer but in the absence of PDS. The mixture was then diluted ten times with infection medium (DMEM plus 2% fetal bovine serum) and transferred onto the cells on the coverslips. The respective virus was allowed to enter the cells, but not to uncoat by the presence of 25 mM NH_4_Cl for 1 h at 34 °C; ammonium ions neutralize the endosomal acidic pH preventing the structural changes necessary for the release of viral RNA and infection. NH_4_Cl was washed away, setting the time for synchronized productive infection. At 8 h post-infection, the cells were washed, fixed, permeabilized, and blocked with goat serum. The RV-A2-infected sample was incubated with monoclonal antibody 8F5, diluted to 10 µg/mL in goat serum, and the RV-B14 and RV-A89 infected samples with specific rabbit antisera diluted to 1:500 in goat serum. For detection, secondary antibodies labeled with Alexa fluor (Life Technologies, Carlsbad, CA, USA) were used at 1 μg/mL, followed by extensive washing. The slides were mounted, and fluorescence microscopy images were recorded on a TissueFAXS automated microscope (TissueGnostics, Vienna, Austria) as in Ganjian et al. [14]. The presence of PDS diminished the number of virus-positive cells in all cases by about 70% compared with cells identically infected with the respective untreated virus. In the continuous presence of the uncoating inhibitor NH_4_Cl, virus production was reduced to about 99% in each instance (Figure 2b).

Finally, we induced release of neutrophil ETs with phorbol 12-myristate 13-acetate acetate (PMA) from isolated neutrophils and observed significant capturing of added RV-A2 particles on elastase-decorated chromatin-based nets (Appendix A and Movie S1). Taken together, we conclude that the PDS fibrils might act similarly to ETs by trapping the virus and preventing it from binding and being taken up by the cells. However, more work is required to find out whether the observed viral trapping could indeed be used in antiviral therapy. It had been shown that Tris, called ‘Tromethamine’ in the below citation, administered through nebulization alleviated respiratory problems resulting from cystic fibrosis [15]; adding the minimally toxic PDS to such solutions might be a way of application.

## Figures and Tables

**Figure 1 viruses-12-00723-f001:**
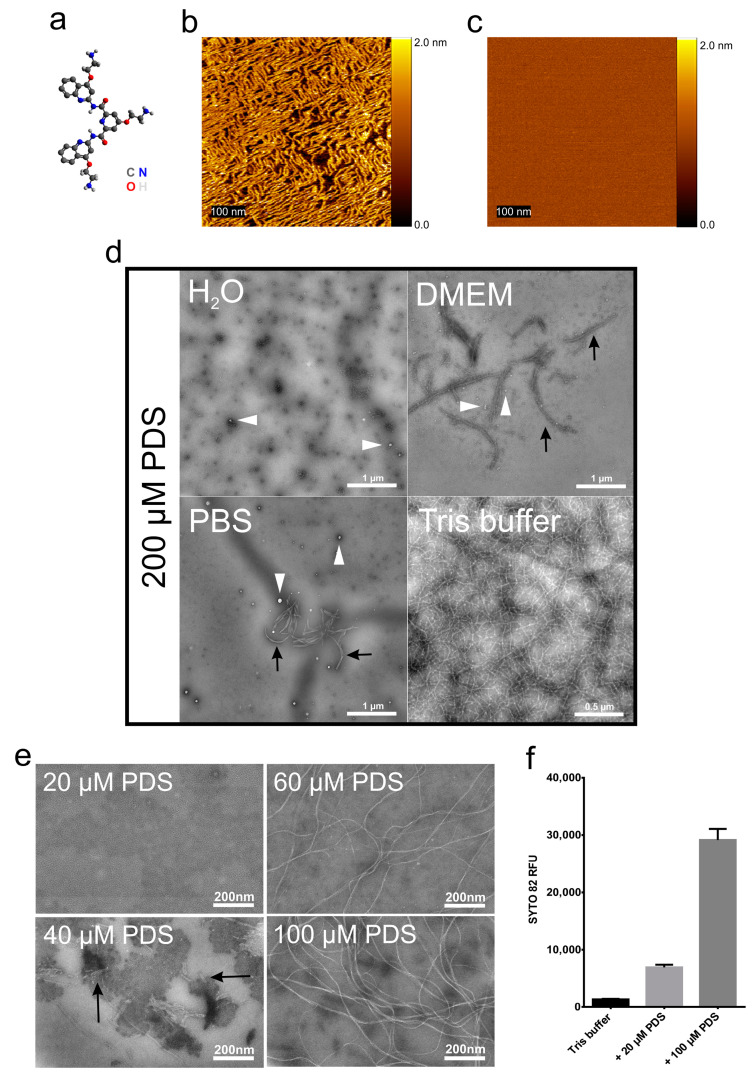
Formation of PDS fibrils under various conditions. (**a**) The structural formula of pyridostatin; for atom colour codes, see lower right. (**b**) PDS (4 µM) in water was applied to freshly cleaved mica, incubated for 5 min, and viewed by AFM. (**c**) As (**b**) but with 4 µM PDS in PBS instead of water. (**d**) PDS dissolved in the solutions indicated at 200 µM was applied onto EM grids, stained with 2% phosphotungstic acid (pH 7.4), and imaged by TEM. The white arrowheads point to small amorphous aggregates, the black arrows to ‘protofibrils’. (**e**) PDS dissolved in Tris buffer at the concentrations indicated was applied to an EM grid, stained with 2% phosphotungstic acid (pH 7.4), and imaged by TEM. (**f**) SYTO82 at 5 µM in Tris buffer +/- PDS at the final concentrations indicated was transferred to a quartz cuvette and the fluorescence signal acquired at room temperature (excitation 541 nm ⁄ emission 560 nm). Data were analysed by one-way ANOVA with Tukey’s multiple comparisons, revealing that the values are significantly different from each other (*p* < 0.05). *n* = 3.

**Figure 2 viruses-12-00723-f002:**
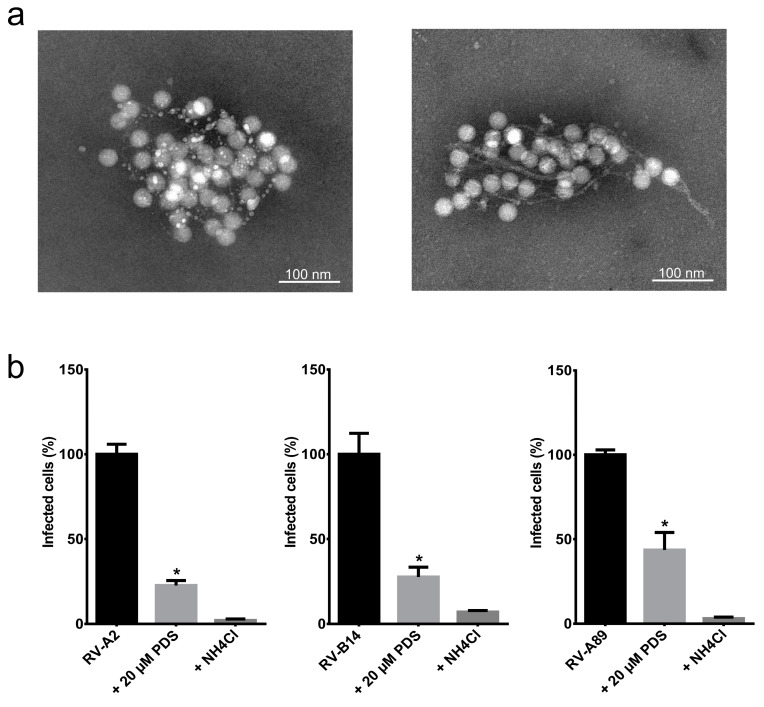
PDS fibrils entrap various RVs. (**a**) RV-A2 (1 µg/mL) was mixed with PDS at 20 µM in Tris buffer, transferred onto EM grids, stained with 2% phosphotungstic acid (pH 7.4), and viewed by TEM. (**b**) HeLa cells, 80% confluent, were either infected with RV-A2, RV-B14, or RV-A89, respectively (controls), or pre-incubated with 20 µM PDS in Tris buffer for 30 min on ice prior to infection as above. The samples were diluted 10 times in infection medium plus 25 mM NH_4_Cl and added to the cells. One hour post-challenge, the medium was replaced with fresh infection medium without NH_4_Cl to initiate uncoating. As a second control, NH_4_Cl was maintained throughout the experiment. At 8 h post-infection, the cells were prepared for immunofluorescence, and the number of cells producing viral antigen, indicating infection, was determined in a TissueFAXS. The average and standard error of the mean of infected cells from three independent assays were plotted. The figure was prepared and the significance levels determined by using GraphPad Prism 6.0 using one-way ANOVA. * *p* < 0.0001 vs. RVs without PDS.

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
