# Peer review of "Catching Common Cold Virus with a Net: Pyridostatin Forms Filaments in Tris Buffer That Trap Viruses—A Novel Antiviral Strategy?"

_viruses, 2020, doi:10.3390/v12070723_

Round 1
Reviewer 1 Report
Real-Hohn et al highlight a very interesting potential aspect of viral trapping by extracellular traps (ET) and synthetic ET trap mimics (pyridostatin) in their Brief Report entitled "Catching common cold virus with a net: Pyridostatin forms filaments in TRIS-buffer trapping viruses. A novel antiviral strategy?". The concept regarding "construction" of viral trapping lattices from non-chromatin organic compounds such as pyridostatin is novel and important. Furthermore, applications of this technology may improve viral trapping now seen in chromatin-based ETs without the potential for inflammatory tissue damage which sometimes complicates dysregulated ET formation in inflammatory syndromes. The manuscript is well written although some further review for English-language nuance would make it even better. There are, however, several issues that will need to be addressed by the authors.
Major:
- The authors clearly outline a progression of experiments from observation of pyridostatin fibrils, to optimization of fibril formation at different concentrations and buffers (100 mcM; TRIS-buffer), and finally, to testing in a physiologic model of viral infection with RV-A2, RV-B14, and RV-A89 viruses. This represents a strength. However, given the focus of comparing pyridostatin fibrils to chromatin-based fibrils associated with ETs, the authors should include a study using chromatin-based fibrils from human leukocytes as a control for viral trapping. Culture of neutrophils with LPS or PMA for 1 - 2 hours would generate ETs that could be used for this purpose.
- Figure 1f - No statistical analysis is outlined in the figure or figure legend. The sample size for the groups in figure 1f should be listed in the figure legend.
- Figure 1b,c - The concentration of pyridostatin between b and c is different. The authors should explain why this is so. Could the lack of fibrils in c be due to the concentration difference?
Minor:
- The reference list is off by one. The reference noted as number 1 in the text is actually number 2 in the reference list.
- In the abstract, the authors state that "Common cold viruses promote this process (Line #18) ...". I do not see any experiments that suggest that cold viruses promote the organization of pyridostatin into fibrils.
- The statement of findings in room temperature experiments listed in lines 46-48 should be more clearly developed in the discussion. In particular, how do the findings presented in this Brief Report fit into the context of viral trapping if they are only active at very low temperatures?
- Figure 2a - Would arrowheads improve communication for these images? Fibrils and virus arrowheads?
Author Response
Major comments:
1) A study of the interaction of RV with chromatin-based fibrils released from PMA-activated neutrophils has now been included as Supplementary Material comprising Supplementary Figure S1 and Supplementary Movie S1; we refer to it at the appropriate location in the main text of the manuscript:
(L33-L35) However, neutrophil ETs also inhibit virus infection [4]. “We here demonstrate, via immunolabeling, that a rhinovirus interacts with them (Supplementary Figure S1 and Supplementary Movie S1).”
(L142-L144) Finally, we induced the release of neutrophil ETs with phorbol 12-myristate 13-acetate acetate (PMA) and found significant capturing of RV-A2 particles on elastase-decorated chromatin-based nets (Supplementary Figure S1 and Supplementary Movie S1).
2) The following information was added to the legend of Figure 1f (L70-L71): Data were analysed by one-way ANOVA with Tukey's multiple comparisons, revealing that the values are significantly different from each other (p < 0.05). N = 3.
3) Definitely, we mistakenly gave a wrong value referring to the PDS concentration in Figure 1b. The correct value is 4 µM. This was corrected in the text in L55 and L62. Thank you for drawing our attention to this!
Minor comments:
1) This has been adjusted.
2) This statement refers to the possibility of the virus particles promoting fibril nucleation, as mentioned in the text (L106-L108): We changed the text as to indicate the possibility but not a fact: “TEM observation suggested that virus particles induced the formation of fibres as they were already observed at 20 µM PDS in their presence but only at 200 µM in their absence (Fig 2a).”
3) The fibres appeared regardless of carrying out the experiments at 4 °C or 25 °C. We finally settled with the lower temperature to exclude diffusion of PDS into the capsid. This can cause additional effects via its interaction with the viral genome. This is now explained in the following sentence (L50-L52): “Inhibition of infection upon incubation at room temperature was, however, independent from the buffer composition and only due to the above G4-stabilization (manuscript in preparation)”.
4) We believe that the addition of arrowheads will not substantially improve the information provided by this image. We hope that leaving it unchanged is acceptable for reviewer 1 as it was also not an issue for the second reviewer.
Reviewer 2 Report
This is an interesting manuscript. The results are clear and support the author’s conclusions. I had a few editing comments (below)
- Title. Catching common cold virus with a net: Pyridostatin 2 forms filaments in TRIS-buffer trapping viruses. A novel antiviral strategy?
I might have put …..Catching common cold virus with a net: Pyridostatin 2 forms filaments in TRIS-buffer that traps viruses. A novel antiviral strategy?
- Line 12. The neutrophil extracellular trap (ET) is a eukaryotic host defence machinery operating via capturing and concentrating pathogens in a filamentous network manufactured by neutrophils and made of DNA, histones and many other components.
Suggest…….The neutrophil extracellular trap (ET) is a eukaryotic host defence machinery that operates by capturing and concentrating pathogens in a filamentous network manufactured by neutrophils and made of DNA, histones and many other components.
- Line 46. I think you should add the word infection…….Inhibition of infection upon incubation at room temperature was, however, independent from the buffer and only due to 48 the above G4-stabilization
- Line 96. This sentence is unclear…..”exclude?”……. To test that in a physiologically relevant system, we infected HeLa cells with PDS-treated virus but had to exclude that the fibres damage the cells as natural ETs do (see above).
Author Response
1) Thank you, we changed the title following your suggestion.
2) The suggestion was incorporated in the abstract (L14-L16).
3) The suggested addition to the text was incorporated (L50).
4) Aiming to clarify this the following sentences were changed (L100-L106): “To test that in a physiologically relevant system we measured the infectivity of PDS-treated virus in HeLa cells; we are aware that such an experiment does not take into account that the fibres might damage the cells similarly to natural ETs (see above); if so, the decreased cell survival could be misinterpreted as increased infectivity. To avoid this, we first decreased the PDS concentrations to 20 µM and incubated 1 µg / ml RV-A2 in Tris buffer for 30 min on ice to prevent capsid breathing and thus the interaction of PDS with the viral genome within the protein shell.”
Round 2
Reviewer 1 Report
The authors have responded to my comments adequately and the manuscript is much improved. The supplemental figure data now provided directly address the most important of my major comments.